# A Polymorphism in the TMPRSS2 Gene Increases the Risk of Death in Older Patients Hospitalized with COVID-19

**DOI:** 10.3390/v14112557

**Published:** 2022-11-18

**Authors:** Clara Caldeira de Andrade, Ana Tércia Paulo Silva, Luydson Richardson Silva Vasconcelos, Pablo Rafael Silveira Oliveira, Carlos Dornels Freire de Souza, Anderson da Costa Armstrong, Rodrigo Feliciano do Carmo

**Affiliations:** 1Post-Graduate Program in Biosciences, Universidade Federal do Vale do São Francisco (UNIVASF), Petrolina 56304-917, Brazil; 2Collegiate of Biological Sciences, Universidade Federal do Vale do São Francisco (UNIVASF), Petrolina 56304-917, Brazil; 3Department of Parasitology, Aggeu Magalhães Institute (IAM), Recife 50740-465, Brazil; 4Biology Institute, Federal University of Bahia (UFBA), Salvador 40170-115, Brazil; 5Collegiate of Medicine, Universidade Federal do Vale do São Francisco (UNIVASF), Petrolina 56304-917, Brazil

**Keywords:** coronavirus, mortality, polymorphism, SARS-CoV-2, TMPRSS2

## Abstract

Background: Transmembrane serine protease type 2 (TMPRSS2) and angiotensin-converting enzyme 2 (ACE2) are the main molecules involved in the entry of SARS-CoV-2 into host cells. Changes in TMPRSS2 expression levels caused by single nucleotide polymorphisms (SNPs) may contribute to the outcome of COVID-19. The aim was to investigate the association between TMPRSS2 gene polymorphisms and the risk of death in hospitalized patients with COVID-19. Methods: We included patients with confirmed COVID-19, recruited from two hospitals in northeastern Brazil from August 2020 to July 2021. Two functional polymorphisms (rs2070788 and rs12329760) in TMPRSS2 were evaluated by real-time PCR. The Kaplan–Meier method was used to estimate death. The Cox’s proportional hazards model was used to adjust for potentially confounding factors. Results: A total of 402 patients were followed prospectively. Survival analysis demonstrated that older patients carrying the rs2070788 GG genotype had shorter survival times when compared to those with AG or AA genotypes (*p* = 0.009). In multivariable analysis, the GG genotype was a factor independently associated with the risk of death in older individuals (hazard ratio = 4.03, 95% confidence interval 1.49 to 10.84). Conclusions: The rs2070788 polymorphism in TMPRSS2 increases risk of death four-fold in older patients hospitalized with COVID-19.

## 1. Introduction

The coronavirus disease 2019 (COVID-19), caused by severe acute respiratory syndrome coronavirus 2 (SARS-CoV-2), emerged in China and then spread around the globe, leading to a pandemic scenario and causing millions of deaths [1]. Although most cases present mild symptoms, a smaller number of infected individuals progress to acute respiratory disease with hypoxia, resulting in hospitalization and other complications such as multiple organ failure or death [2]. Age, sex, presence of comorbidities, and viral variants are known factors that may contribute to the risk of complications in COVID-19 [3,4,5,6]. In addition, there is now robust evidence that host genetic factors may influence the susceptibility and clinical course of SARS-CoV-2 infection [7,8].

The transmembrane serine protease type 2 (TMPRSS2) protein plays a key role in coronavirus infections [9,10,11], including SARS-CoV-2 [12]. SARS-CoV-2 enters the host cells by binding to angiotensin-converting enzyme type 2 (ACE2), followed by its priming by TMPRSS2. The interaction between ACE2 and TMPRSS2 with the viral spike protein is essential for fusion with host-membrane and endocytosis [12,13]. Furthermore, evidence suggests that the SARS-CoV-2/TMPRSS2/ACE2 interaction may contribute to an imbalance of the renin–angiotensin system, contributing to a worse clinical outcome, especially in high-risk groups [14].

The TMPRSS2 gene, located at human chromosome 21q22.3, is approximately 44 kb in length and includes 14 exons [15]. In 2015, it was demonstrated for the first time that a single nucleotide polymorphism (SNP) in TMPRSS2 (rs2070788) increases the risk of severe H1N1 influenza [16]. With the COVID-19 pandemic, several studies investigated the potential roles of TMPRSS2 variants in susceptibility to COVID-19 through in silico studies using public genomic databases [17,18,19,20,21]. However, two SNPs, rs2070788 and rs12329760, involved with alterations in TMPRSS2 expression levels, stood out [17,19,21].

The rs2070788 is a G-to-A transition located in intron 11–12 (c.1171 + 587C > T) of TMPRSS2. In a lung expression quantitative trait loci analysis, rs2070788 was correlated to the differential expression of TMPRSS2, with GG genotype carriers showing the highest expression levels [21], probably caused by a linkage disequilibrium with another SNP located in a regulatory region. On the other hand, the rs12329760 is a missense variant that causes a change in the amino acid valine at position 160 to methionine, resulting in a moderately less catalytically active version of TMPRSS2, which is less able to self-cleave and prime the SARS-CoV-2 spike protein [22].

Studies evaluating the role of SNPs in TMPRSS2 with the risk of death in hospitalized patients with COVID-19 are scarce, usually with a small sample size, retrospective, and performed in Asian populations [23,24]. In the present study, we investigated the role of rs2070788 and rs12329760 polymorphisms in TMPRSS2 with the risk of death in patients with COVID-19 followed up in two referral centers in the Northeast Region of Brazil.

## 2. Materials and Methods

### 2.1. Study Participants

The study included patients over 18 years of age, with SARS-CoV-2 infection confirmed by molecular (RT-qPCR) or serological test, who were hospitalized between August 2020 and July 2021, at the University Hospital of the Universidade Federal do Vale do São Francisco (HU-UNIVASF/EBSERH, acronym in Portuguese) and the Field Hospital of the Municipality of Petrolina, which are both reference centers for treatment of COVID-19 in the Vale do São Francisco Region, Petrolina, Northeast Brazil. Follow-up ended at the date of death or discharge. Individuals without information about the date of symptom onset and those transferred to other hospitals were excluded from the study.

All participants or their relatives signed the institutional consent letter. The study was conducted following the Declaration of Helsinki and approved by the Ethics Committee of the Hospital das Clínicas of the Federal University of Pernambuco (HC/UFPE, acronym in Portuguese) under number CAAE: 38196620.0.0000.8807.

### 2.2. DNA Extraction and Genotyping

A 4 mL peripheral blood sample was collected in an EDTA tube (Vacuette K3EDTA tube, Greiner Bio-One, Kremsmünster, Austria) within 24 h of hospital admission. Genomic DNA was obtained from peripheral blood samples, using a commercial extraction kit (ReliaPrep Blood gDNA Miniprep System, Promega, Madison, WI, USA), following manufacturer’s instructions. DNA samples were eluted to a final volume of 100 μL and quantified using NanoDrop OneC spectrophotometer (Thermo Scientific, Waltham, MA, USA). The genetic material obtained was stored in a freezer at −20 °C until genotyping was performed.

Two polymorphisms in TMPRSS2 (rs2305619 and rs1840680) were selected based on previous evidence of association with respiratory viruses and in silico functional prediction analyses [16,21,22]. Polymorphisms were determined by real-time PCR (QuantStudio 5, Thermo Fisher), using pre-designed TaqMan assays (Assay ID C__22275654_10 and C__12069244_10 for rs2305619 and rs1840680, respectively). The reaction mixture contained 5 μL of GoTaq Probe qPCR Master Mix (2×) (Promega, Madison, WI, USA), 0.25 μL of TaqMan assay, 2.75 μL of sterile water (DNase and RNase free), and 2.0 μL of genomic DNA, in a final volume of 10 μL. The qPCR thermal conditions were as follows: 50 °C for 2 min, 95 °C for 10 min, 40 cycles at 95 °C for 15 s, and 60 °C for 1 min. Allelic discrimination plots are shown in Appendix A.

### 2.3. Statistical Analysis

Considering the influence of age on TMPRSS2 expression levels [25], a cutoff point was set at the median of the sample (≥55 years of age) in the analyses of this study.

Comparisons of continuous variables were performed using Student’s *t*-test. For categorical variables, Pearson’s chi-square test and Fisher’s exact test were used. The Kaplan–Meier method was used to estimate the influence of polymorphisms in TMPRSS2 on the risk of death. Time was considered from symptom onset to outcome. The log-rank test was used for genotype comparison. Variables previously associated with the risk of death, such as age and sex [6], as well as comorbidities with a *p* value < 0.05 in the univariate analysis were selected for inclusion in the Cox’s proportional hazards model, using a stepwise backward procedure. Values of *p* < 0.05 were considered significant.

Data were analyzed using SPSS Statistics v.22.0 software (SPSS, Inc., Chicago, IL, USA). Graphpad Prism version 8.0 (Graphpad, San Diego, CA, USA) was used to construct the graphs.

## 3. Results

### 3.1. Baseline Characteristics

A total of 402 patients were enrolled in the current study (245 men and 157 women). The mean time from onset of symptoms to the end of follow-up was 18.5 ± 11.0 days. During follow-up, 61 (15.1%) patients died. Table 1 describes the baseline characteristics of the patients included in the study according to the outcome. The factors of age, diabetes, hypertension, chronic heart disease, chronic kidney disease, and obesity were significantly more prevalent in individuals who died in comparison to those who survived (*p* < 0.05).

### 3.2. Influence of the TMPRSS2 Polymorphisms on the Risk of Death

The TMPRSS2 rs2070788 and rs12329760 genotype distributions did not significantly differ from the Hardy–Weinberg equilibrium expectations (*p* = 0.10 and *p* = 0.34, respectively). A higher G allele frequency of rs2070788 was observed in individuals who died compared to survivors (*p* = 0.019). In the genotype analysis, there was a higher frequency of the GG genotype in the death group; however, the difference was not statistically significant (*p* = 0.08). No association was observed between the rs12329760 polymorphism and the occurrence of death (Table 2). The minor allele frequency (MAF) of the two polymorphisms in different populations available in public databases [26,27] is presented in Appendix A.

The expression of TMPRSS2 in mouse and human lung tissue is age-related [25]. Based on this, we decided to perform the same analysis of the rs2070788 and rs12329760 polymorphisms in a group of individuals aged 55 years or older. A total of 175 older individuals were analyzed, of whom 39 (22.2%) died. A significant association between the rs2070788 G allele and the occurrence of death was observed (*p* = 0.008). Older patients carrying the GG genotype had a higher incidence of death (*p* = 0.01). No associations with the rs12329760 polymorphism were observed (Table 2).

Using the Kaplan–Meier method (Figure 1), we demonstrated that older individuals carrying the GG genotype (rs2070788) had a shorter survival time from symptom onset when compared to those with the AG or AA genotype (mean survival time in days, GG: 28.7, AG: 36.9, AA: 48.6, log-rank = 0.009). There was no difference in survival time between the TMPRSS2 rs12329760 genotypes (Figure 1).

### 3.3. Multivariable Analysis to Identify Predictors of Death in Older Individuals

To verify whether the TMPRSS2 rs2070788 polymorphism can predict the risk of death in older individuals with COVID-19 independent of other confounding factors, we performed a Cox’s proportional hazards regression model including potentially confounding factors associated with COVID-19 outcome. The following variables were included: TMPRSS2 rs2070788 genotypes, sex, age, diabetes, hypertension, chronic heart disease, chronic kidney disease, and obesity.

In the multivariable analysis, carrying the GG genotype increased the risk of death in older individuals hospitalized for COVID-19 four-fold, independent of other factors (hazard ratio = 4.03, 95% confidence interval 1.49 to 10.84, *p* = 0.006). Chronic heart disease (*p* = 0.024), chronic kidney disease (*p* < 0.0001), and obesity (*p* = 0.053) also remained in the final model (Table 3).

## 4. Discussion

The current study assessed the influence of genetic polymorphisms modulating TMPRSS2 expression on the risk of death in 402 hospitalized patients with COVID-19. We demonstrated for the first time that older patients carrying the rs2070788 GG genotype, which is associated with high TMPRSS2 expression in human lung tissue [16,21], had a higher risk of death by COVID-19.

Since the first evidence demonstrating the importance of TMPRSS2 in SARS-CoV-2 entry [12], several studies have been developed in order to investigate the association between genetic polymorphisms in TMPRSS2 and the risk of COVID-19, using genomic databases [17,18,19,20,21]. Among the various SNPs in the TMPRSS2 gene, some have shown promise, as they are associated with changes in TMPRSS2 expression levels, including rs2070788 and rs12329760 [16,21,22].

The first study to demonstrate the role of the TMPRSS2 rs2070788 polymorphism in the pathophysiology of a viral respiratory infection was performed on Asian patients with H1N1 influenza infection. This study demonstrated that the rs2070788 GG genotype conferred a more than two-fold increased risk of severe H1N1 influenza. They suggested that rs2070788 was in linkage disequilibrium with another variant, rs383510, located in a regulatory region of the gene [16]. In agreement with this finding, a recent Dutch study of 188 adult hospitalized patients demonstrated a protective effect of the rs2070788 AA genotype on COVID-19 severity [28]. On the other hand, a German study evaluating 239 patients with COVID-19 found no association between the TMPRSS2 rs2070788, rs12329760, or rs383510 polymorphisms and COVID-19 severity. Nonetheless, they did show an association between the rs383510 variant and the risk of SARS-CoV-2 infection when comparing infected and uninfected individuals [29].

Our findings support the hypothesis that TMPRSS2 may contribute to the outcome of COVID-19 in older individuals. An in silico study using several large genomic databases demonstrated that the G allele of rs2070788 is associated with elevated TMPRSS2 expression in lung tissue, and that the frequency of this allele is higher in European and American populations than in Asian populations [21]. ACE2, in addition to its role in SARS-CoV-2 entry [12], has an important protective function against acute cardiovascular and lung damage [14]. During SARS-CoV infection, membrane-bound levels of active ACE2 can be decreased through TMPRSS2-mediated cleavage or by the infection itself. This can cause an imbalance in the renin–angiotensin system causing seriously damaging effects in pulmonary and cardiac tissues, especially in individuals with comorbidities [14,30,31]. The same is presumed for SARS-CoV-2, given that SARS-CoV and SARS-CoV-2 engage ACE2 in the same manner [32]. Therefore, in older individuals, where TMPRSS2 levels are known to be higher [25], carriage of the rs2070788 GG genotype in TMPRSS2 could exacerbate ACE2 cleavage and be decisive for clinical outcome.

In the present study, we found no association between the rs12329760 polymorphism and the risk of death due to COVID-19. This variant has been widely studied in the context of COVID-19 due to its functional effect that causes a decrease in the proteolytic activity of TMPRSS2 [22]. It is possible that the damaging effects of TMPRSS2 on the renin–angiotensin system are more related to variants that alter the expression levels of TMPRSS2, rather than structural changes in the protein. This could explain the lack of association between this mutation and risk of death in our study. Studies evaluating the role of rs12329760 in the outcome of COVID-19 are scarce. An Iranian study evaluating 288 hospitalized patients with COVID-19 also found no association between the rs12329760 variant and the risk of death [24]. Other studies have shown divergent results and are focused on assessing the role of rs12329760 in severity or susceptibility to SARS-CoV-2 infection [22,23,29,33,34,35,36,37], and therefore cannot be directly compared with our results. The following factors may contribute to the divergent results in these studies: lack of statistical power due to small sample sizes, different severity classification criteria, and differences in the genetic background of the populations analyzed. 

The limitations to this study include lack of information on viral aspects, such as viral load and variants. Although we do not have data at the individual level, it is important to note that the patient recruitment period of the present study coincides with the circulation of the gamma variant in Brazil [38].

In summary, our results, using survival analysis, demonstrated for the first time that older individuals carrying the rs2070788 GG genotype in the TMPRSS2 gene, previously associated with elevated expression levels in lung tissue, have a higher risk of COVID-19 death. This is the first study evaluating the role of polymorphisms in TMPRSS2 in the context of COVID-19 in a Latin American population. These findings pave the way for future studies in other populations, focusing especially on individuals of advanced age.

## Figures and Tables

**Figure 1 viruses-14-02557-f001:**
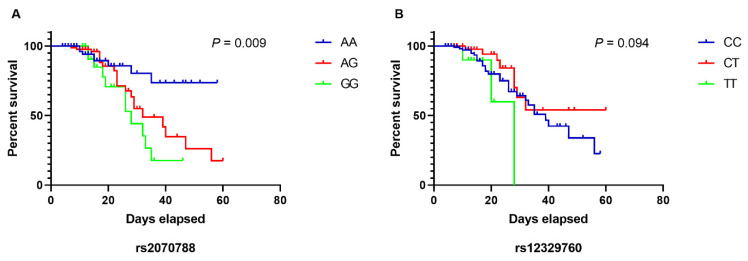
Influence of the transmembrane serine protease type 2 (TMPRSS2) polymorphisms on the risk of death in older individuals hospitalized with COVID-19 according to Kaplan–Meier method. (**A**) rs2070788, mean time (days) GG: 28.7, AG: 36.9, AA: 48.6, log-rank = 0.009. (**B**) rs12329760, mean time (days) TT: 23.8, CT: 44.4, TT: 38.3, log-rank = 0.094.

**Table 1 viruses-14-02557-t001:** Baseline characteristics of the patients included in the study according to the outcome.

Variables	Survived (*n* = 341)	Deceased (*n* = 61)	*p* Value
Age (mean ± SD)	51.4 ± 14.3	60.4 ± 16.7	0.001
Sex			
Male (*n*, %)	207 (60.7)	38 (62.3)	0.887
Female (*n*, %)	134 (39.3)	23 (37.7)	
Comorbidities			
Diabetes (*n*, %)	92 (27.1)	27 (44.3)	0.009
Hypertension (*n*, %)	154 (45.3)	37 (60.7)	0.036
Chronic heart disease (*n*, %)	6 (1.8)	5 (8.2)	0.015
Chronic kidney disease (*n*, %)	5 (1.5)	7 (11.5)	0.001
Chronic liver disease (*n*, %)	0 (0.0)	1 (1.6)	0.152
Malignancy (*n*, %)	3 (0.9)	1 (1.6)	0.485
Asthma (*n*, %)	11 (3.2)	1 (1.6)	1.000
COPD (*n*, %)	12 (3.5)	5 (8.2)	0.156
Rheumatic disease (*n*, %)	3 (0.9)	0 (0.0)	1.000
Obesity (*n*, %)	78 (22.9)	23 (37.7)	0.024

COPD: chronic obstructive pulmonary disease.

**Table 2 viruses-14-02557-t002:** Allelic and genotypic distribution of rs2070788 and rs12329760 variants of TMPRSS2 according to outcome.

	All (*n =* 402)			Older Individuals (*n =* 175)		
TMPRSS2 (rs2070788)	Survived (*n =* 341)	Deceased (*n =* 61)	*p* Value	Survived (*n =* 136)	Deceased (*n =* 39)	*p* Value
*Alleles*	***n* (%)**	***n* (%)**		***n* (%)**	***n* (%)**	
A	407 (59.6)	59 (48.3)	0.019	171 (62.8)	36 (46.1)	0.008
G	275 (40.3)	63 (51.6)		101 (37.1)	42 (53.8)	
*Genotypes*	***n* (%)**	***n* (%)**		***n* (%)**	***n* (%)**	
AA	128 (37.5)	15 (24.6)		51 (37.5)	8 (20.5)	
AG	151 (44.3)	29 (47.5)	0.081	69 (50.7)	20 (51.3)	0.019
GG	62 (18.2)	17 (27.9)		16 (11.8)	11 (28.2)	
**TMPRSS2 (rs12329760)**						
*Alleles*	***n* (%)**	***n* (%)**		***n* (%)**	***n* (%)**	
C	537 (78.7)	102 (83.6)	0.220	212 (77.9)	64 (82.0)	0.433
T	145 (21.2)	20 (16.3)		60 (22.0)	14 (17.9)	
*Genotypes*	***n* (%)**	***n* (%)**		***n* (%)**	***n* (%)**	
CC	213 (62.5)	44 (72.1)		84 (61.8)	28 (71.8)	
CT	111 (32.6)	14 (23.0)	0.318	44 (32.4)	8 (20.5)	0.356
TT	17 (4.9)	3 (5.0)		8 (5.9)	3 (7.7)	

TMPRSS2: transmembrane serine protease type 2.

**Table 3 viruses-14-02557-t003:** Characteristics associated with the risk of death according to Cox’s proportional hazards model.

Variables	HR (95% CI)	*p* Value
**TMPRSS2 (rs2070788)**		
AA	Reference	-
AG	2.15 (0.93–4.97)	0.073
GG	4.03 (1.49–10.84)	0.006
**Comorbidities**		
Chronic heart disease	3.19 (1.16–8.72)	0.024
Chronic kidney disease	5.12 (2.10–12.46)	<0.0001
Obesity	2.08 (0.98–4.37)	0.053

Variables entered in the initial model: TMPRSS2 rs2070788, sex, age, diabetes, hypertension, chronic heart disease, chronic kidney disease, and obesity. CI: confidence interval; HR: hazard ratio; TMPRSS2: transmembrane serine protease type 2.

## Data Availability

The data presented in this manuscript will be available from the corresponding author upon reasonable request.

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
