# Peer review of "A Polymorphism in the TMPRSS2 Gene Increases the Risk of Death in Older Patients Hospitalized with COVID-19"

_viruses, 2022, doi:10.3390/v14112557_

Round 1

Reviewer 1 Report

In the manuscript entitled “A polymorphism in the TMPRSS2 gene increases the risk of death in older patients hospitalized with COVID-19”, the authors got good results that the rs2070788 TMPRSS2 polymorphism are associated with the risk of death in older patients hospitalized with COVID-19. It is interesting study using many dead patients, however there are still some scientific criticisms need to be addressed by the authors to strengthen the manuscript.

1.     The authors should explain in more detail which laboratory or equipment was used to experiment with whole blood samples of COVID-19 patients.

2.     The authors only performed SNP Genotyping using RT-PCR. The RT-PCR experimental method should be described in more detail. In addition, DNA sequencing and RT-PCR results for patients with three genotypes should be provided as Supplementary data.

3.     What variant type of COVID-19 virus were the patients infected with?  Ex) delta, Epsilon, Omicron…

4.     I recommend adding data from healthy Brazilians for two SNPs and comparing them with HWE analysis.

5.     I recommend analyzing and comparing the haplotype for two SNPs in these data.

Line 149, TT: 38.3 à CC: 38.3

Author Response

Dear Reviewer, thank you for your contributions and time spent reading our manuscript. We hope that this new version is suitable for publication. Best wishes.

  1. The authors should explain in more detail which laboratory or equipment was used to experiment with whole blood samples of COVID-19 patients.

A: Dear reviewer, thank you for your comment. We have added more details about sample collection and processing in subtopic 2.2 DNA extraction and genotyping.

  1. The authors only performed SNP Genotyping using RT-PCR. The RT-PCR experimental method should be described in more detail. In addition, DNA sequencing and RT-PCR results for patients with three genotypes should be provided as Supplementary data.

A: Dear reviewer, thanks for the suggestion. We have added more details about the genotyping assay in the Methods section. In addition, we have added a figure in the supplementary materials, as suggested. Because these are validated TaqMan assays and are highly specific for the variants studied, we did not perform sequencing of the samples.

  1. What variant type of COVID-19 virus were the patients infected with? Ex) delta, Epsilon, Omicron…

A: Dear reviewer, unfortunately we have no data on what type of variant the patients were infected with. That is a limitation of this study, and we recognize that. We decided to add the following sentence at the end of the discussion:

“The limitations to this study include lack of information on viral aspects, such as viral load and variants. Although we do not have data at the individual level, it is important to note that the patient recruitment period of the present study coincides with the circula-tion of the gamma variant in Brazil [36].”

  1. I recommend adding data from healthy Brazilians for two SNPs and comparing them with HWE analysis.

A: Dear reviewer, thank you for your comment. We have added a supplementary table with the minor allele frequency available in public databases in different populations, including a Brazilian population from southeastern Brazil. Brazil is a country of continental dimensions and with a highly mixed population that varies according to region. In the present study we included a population from the northeast of Brazil, and unfortunately, we do not have data from a healthy population from the same region. Also, since the databases provide only allele frequency information, we cannot calculate HWE in these populations. We have added a new sentence in section 3.2 of the results.

  1. I recommend analyzing and comparing the haplotype for two SNPs in these data.

A: Dear reviewer, the two SNPs included in this study are not in linkage disequilibrium (R2=0.065), so haplotype analysis cannot be performed.

Reviewer 2 Report

Current work described the importance of rs2070788 polymorphism in decreased survival of COVID-19 patients.

While it is true that it is the first study evaluating the role of polymorphisms in TMPRSS2 in the context of COVID-19 in a Latin American population, how did the authors selected only the said two polymorphisim? What are the other few polymorphisms in the same gene? What could be their effect on the morbidity?

Also which particular comorbid patients were more vulnerable? 

What could be the effect if this polymorphisim in the protein level of TMPRSS2? In terms of structure? Binding affinity etc.?

Author Response

Dear Reviewer, thank you for your contributions and time spent reading our manuscript. We hope that this new version is suitable for publication. Best wishes.

Current work described the importance of rs2070788 polymorphism in decreased survival of COVID-19 patients.

While it is true that it is the first study evaluating the role of polymorphisms in TMPRSS2 in the context of COVID-19 in a Latin American population, how did the authors selected only the said two polymorphisim? What are the other few polymorphisms in the same gene? What could be their effect on the morbidity?

A: Dear reviewer, thank you for your comment. As said in the introduction of the article, the two SNPs evaluated herein are the most studied in the literature, since they are related to changes in protein expression levels. In the context of COVID-19 the rs12329760 is the most studied. In a European study evaluating 377 SNPs in the TMPRSS2 gene, rs12329760 was the only one that was shown to be relevant in the context of the clinical course of the disease (David et al., 2022, doi:10.1016/j.retram.2022.103333).

In the fourth paragraph of our manuscript it states:

“The rs2070788 is a G-to-A transition located in intron 11–12 (c.1171+587C>T) of TMPRSS2. In a lung expression quantitative trait loci analysis, rs2070788 was correlated to differential expression of TMPRSS2, with GG genotype carriers showing the highest expression levels [21], probably caused by a linkage disequilibrium with another SNP located in a regulatory region. On the other hand, the rs12329760 is a missense variant that causes a change in the amino acid valine at position 160 to methionine, resulting in a moderately less catalytically active of TMPRSS2, which is less able to self-cleave and prime the SARS-CoV-2 spike protein [22].”

Also which particular comorbid patients were more vulnerable?

A: In table 3 in the multivariate analysis it was observed that individuals with chronic heart disease, chronic kidney disease and obesity were more vulnerable.

What could be the effect if this polymorphisim in the protein level of TMPRSS2? In terms of structure? Binding affinity etc.?

A: In the fourth paragraph of the introduction it says:

“The rs2070788 is a G-to-A transition located in intron 11–12 (c.1171+587C>T) of TMPRSS2. In a lung expression quantitative trait loci analysis, rs2070788 was correlated to differential expression of TMPRSS2, with GG genotype carriers showing the highest expression levels [21], probably caused by a linkage disequilibrium with another SNP located in a regulatory region. On the other hand, the rs12329760 is a missense variant that causes a change in the amino acid valine at position 160 to methionine, resulting in a moderately less catalytically active of TMPRSS2, which is less able to self-cleave and prime the SARS-CoV-2 spike protein [22].”

Round 2

Reviewer 1 Report

The authors answered my questions well and this revised manuscript was satisfactorily modified for my comments. Thus, I am recommending acceptance for publication in the Viruese.